# Feasibility, Reliability, and Safety of Remote Five Times Sit to Stand Test in Patients with Gastrointestinal Cancer

**DOI:** 10.3390/cancers15092434

**Published:** 2023-04-24

**Authors:** Daniel Steffens, Natasha C. Pocovi, Jenna Bartyn, Kim Delbaere, Mark J. Hancock, Cherry Koh, Linda Denehy, Kimberley S. van Schooten, Michael Solomon

**Affiliations:** 1Surgical Outcomes Research Centre (SOuRCe), Royal Prince Alfred Hospital (RPAH), Sydney, NSW 2006, Australia; 2Faculty of Medicine and Health, Central Clinical School, The University of Sydney, Sydney, NSW 2042, Australia; 3Faculty of Medicine, Health and Human Sciences, Macquarie University, Sydney, NSW 2109, Australia; 4Neuroscience Research Australia, Randwick, Sydney, NSW 2031, Australia; 5School of Population Health, University of New South Wales, Kensington, NSW 2052, Australia; 6Department of Colorectal Surgery, Royal Prince Alfred Hospital (RPAH), Sydney, NSW 2006, Australia; 7Department of Physiotherapy, Faculty of Medicine Dentistry and Health Sciences, The University of Melbourne, Melbourne, VIC 3010, Australia; 8Department of Health Services Research, Allied Health, Peter MacCallum Cancer Centre, Melbourne, VIC 3000, Australia

**Keywords:** five times sit to stand test, remote assessment, cancer, reliability, cross-sectional design

## Abstract

**Simple Summary:**

The five times sit to stand (5STS) test is widely used to measure functional lower extremity strength. However, the psychometric properties of the 5STS test when performed remotely is unknown. This study determined the feasibility, reliability, and safety of the remote five times sit to stand test (5STS) in 37 patients scheduled to undergo gastrointestinal cancer surgery. Participants completed the 5STS test both face-to-face and remotely, with the order randomised. The study provides supporting evidence that the remote 5STS test is feasible, reliable, and safe in patients with gastrointestinal cancer and can be used in both clinical and research settings.

**Abstract:**

**Background:** To determine the feasibility, reliability, and safety of the remote five times sit to stand test (5STS) test in patients with gastrointestinal cancer. **Methods:** Consecutive adult patients undergoing surgical treatment for lower gastrointestinal cancer at a major referral hospital in Sydney between July and November 2022 were included. Participants completed the 5STS test both face-to-face and remotely, with the order randomised. Outcomes included measures of feasibility, reliability, and safety. **Results:** Of fifty-five patients identified, seventeen (30.9%) were not interested, one (1.8%) had no internet coverage, and thirty-seven (67.3%) consented and completed both 5STS tests. The mean (SD) time taken to complete the face-to-face and remote 5STS tests was 9.1 (2.4) and 9.5 (2.3) seconds, respectively. Remote collection by telehealth was feasible, with only two participants (5.4%) having connectivity issues at the start of the remote assessment, but not interfering with the tests. The remote 5STS test showed excellent reliability (ICC = 0.957), with limits of agreement within acceptable ranges and no significant systematic errors observed. No adverse events were observed within either test environment. **Conclusions:** Remote 5STS for the assessment of functional lower extremity strength in gastrointestinal cancer patients is feasible, reliable, and safe, and can be used in clinical and research settings.

## 1. Introduction

For cancer patients scheduled for major surgery, the preoperative period provides an opportunity for the identification of modifiable medical, physical, nutritional, and psychological risk factors that may associate with postoperative complications [1]. Patients scheduled for major elective surgical procedures will generally attend a preoperative clinic and undergo a variety of assessments 2–6 weeks before their surgery. However, during the Coronavirus Disease 2019 (COVID-19) pandemic, most of the preoperative assessments were limited to appointments via telehealth, to protect vulnerable cancer patients, clinicians, community, and to control the spread of COVID-19 [2].

Clinical research projects that regularly conduct baseline assessments during the preoperative work-up period were also significantly affected, with most research projects paused or required to rapidly adapt to the challenges associated with the COVID-19 pandemic [3,4]. Some of these challenges, included severe restrictions to face-to-face assessments, hampering the collection of crucial research measures, particularly objective physical outcomes [5]. To minimise the spread of the disease and avoid placing participants and trialists at risk, remote physical assessment was chosen as the preferred contingency plan for many research projects [6].

While remote physical assessments via telehealth [7] enabled researchers to maintain participant involvement in research projects and allowed for the collection of some objective outcome measures, the feasibility, reliability, and safety of specific physical assessments, such as five times sit-to-stand test (5STS), when collected via telehealth is currently not known. At our centre, the gold standard tests for the assessment of physical capacity, including cardiopulmonary exercise testing (CPET) and the six-minute walk test (6MWT), were completely aborted, due to the increased risk of COVID-19 transmission in our laboratory, and particularly for at-risk inpatients within the hospital. Alternatively, the 5STS, Ref. [8] is a simple, quick, and easy test to measure lower extremity muscle strength and is commonly used in various diseases, including cancer patients undergoing surgery [9,10]. When performed face-to-face, the 5STS test has demonstrated excellent psychometric properties of reliability, validity, and responsiveness [11,12]. Therefore, the validation of the remote 5STS test is useful, and can be used beyond the COVID-19 pandemic (e.g., assessing patients rurally residing) as telemedicine is further developed.

Recently, studies have successfully evaluated the reliability of a remote 5STS test [13,14] and other variations, including the 30 s sit to stand (30 s STS) test [15,16]; however, the psychometric properties of the remote 5STS test in cancer patients are currently unknown. The aims of this study were to determine the feasibility, validity, and safety of the remote 5STS test in patients presenting with gastrointestinal cancer.

## 2. Materials and Methods

### 2.1. Study Design and Setting

This cross-sectional observational study included patients seeking surgical treatment for gastrointestinal cancer between July and November 2022 at the Royal Prince Alfred Hospital, Sydney, Australia. This manuscript was written in accordance to the STROBE checklist [17]. Ethics and Governance approvals were obtained from the Sydney Local Health District Ethics Review Committee (Royal Prince Alfred Hospital Zone—Approval number X22-0092/ETH00602), with written informed consent sought from all participants.

### 2.2. Participant Recruitment

Patients were identified by the Royal Prince Alfred Hospital (Sydney, Australia) treating gastrointestinal surgeons and provided with the Participant Information Sheet and Consent Form. Interested participants were contacted by an experienced research officer from the Surgical Outcomes Research Centre (SOuRCe), who provided further information about the study and obtained written consent. Participants aged 18 years and older, presenting with gastrointestinal cancer and seeking surgical treatment at the Royal Prince Alfred Hospital, were included if they had access to a device (e.g., mobile phone, tablet, laptop, or desktop) with internet connectivity, a camera, and audio capabilities. Participants presenting with severe vision and/or hearing impairments, or that were too unwell to perform the 5STS test were excluded.

Those patients who agreed to participate then provided demographic information, and performed the face-to-face and remote 5STS tests.

### 2.3. Demographic Measures

Baseline variables were collected via online or paper questionnaires. These included age, gender, body mass index, country of birth, language spoken at home, caring responsibilities, level of education, employment status, type of cancer, and level of familiarity with technology (i.e., smartphone, computer, tablet, etc.).

Symptoms of pain were collected using the Numerical Pain Rating Scale [18], with scores ranging from 0 to 10, where higher scores indicated the worst pain symptom. Patient distress was assessed using the Distress Thermometer [19], with scores ranging from 0 to 10, where higher scores indicated higher patient distress. Fatigue was measured using the nine-item Fatigue Severity Scale, with scores ranging from 9 to 63, where higher scores indicated more severe fatigue [20]. Measures of physical activity were collected using the International Physical Activity Questionnaire—Short Form (IPAQ-SF) [21]. Participants were categorised as meeting or not meeting at least 150–300 min of moderate-intensity aerobic physical activity, or at least 75–150 min of vigorous-intensity physical activity, or an equivalent combination of moderate- and vigorous-intensity activity throughout the week, per the level of physical activity recommended by the World Health Organisation guidelines.

### 2.4. Five Times Sit to Stand Test

Following baseline data collection, participants were randomly allocated to undertake either the face-to-face or remote 5STS first [8]. Both 5STS assessments were performed at approximately the same time of day, within a 3-day window. The same assessor performed the face-to-face and the remote 5STS assessment for each individual participant.

For the face-to-face 5STS assessment, all participants were assessed using the same chair (44 cm) and clinical assessment room. Participants were instructed to sit with their arms folded across their chest and with their back against a chair. Participants were asked to stand up and sit down as quickly as possible, five times. The time taken in seconds to complete the test was recorded using a stopwatch. A shorter time for completion of the 5STS is indicative of better lower limb strength.

Prior to the remote assessment of 5STS, a trained research officer aided the participants in identifying the most appropriate place to conduct the test, by explaining the test to the participants and instructing them to locate a clear space in their home without furniture or obstacles, to reduce risks. A support person was required to be present during the remote assessment. Participants were asked to identify a chair that was 43–45 cm high to be consistent with the chair used in the face-to-face assessment.

### 2.5. Primary Outcome Measures

The main outcome measures of this study were the feasibility, reliability, and safety of the remote 5STS test, and were defined a priori as:**(i)** ***Feasibility:*** The feasibility of the 5STS test was as the proportion of eligible patients that incurred issues with at home assessment, including inadequate space, chair, or internet connectivity. The remote assessment was considered feasible if the minority (i.e., <20%) of included participants presented with the abovementioned issues.**(ii)** ***Reliability:*** This was measured by comparing the 5STS test scores (i.e., time) between the remote (videoconferencing measurement) and direct assessment (face-to-face measurement), within the same participant. This was performed to explore whether remote physical assessments produce similar scores (i.e., agreement) as the face-to-face assessments.**(iii)** ***Safety:*** Safety was defined by the number of adverse events which occurred during the 5STS tests. A serious adverse event was defined as an event which required medical intervention and results in death, a life-threatening situation, hospitalisation, incapacity, and/or disability. A minor adverse event was defined as an event that requires medical review and resolves without intervention, resulting in no hospitalisation, incapacity, or disability [22].

### 2.6. Sample Size

The sample size of the study was based on the time (seconds) taken to complete the 5STS tests. Thus, 36 participants were required based on an Interclass Correlation Coefficient (ICC) of 0.75, a precision of 0.15, a confidence level of 95%, and a dropout rate of 5%.

### 2.7. Analyses

All study data were recorded in a Research Electronic Data Capture (REDCap, Vanderbilt University, Nashville, TN, USA) database, which was based on a secure server hosted by the Sydney Local Health District [23]. Descriptive statistics were used to characterise the sample, feasibility, and safety outcomes. Categorical data are presented as frequency (percentage) and continuous data as median (interquartile range).

To determine the reliability of the remote 5STS test, the two-way random ICC with single measures were calculated. ICC values range from 0 to 1, where 1 indicates perfect agreement. An ICC ≥ 0.8 was considered high, 0.6 to 0.79 was considered moderate, and <0.6 was considered as having poor validity. The optimal level of agreement was set as 0.8 and the minimum acceptable level was set as 0.7 [24]. To investigate the agreement between the face-to-face and remote physical assessments, a Bland and Altman plot was used [25]. Wilcoxon signed-rank test was used to determine the differences between face-to-face and remote 5STS tests.

The Standard Error of the Measurement (SEM) was used to measure the precision of measurement and the absolute reliability, and was calculated using the following formula: SEM = SD × √1 − ICC [26]. A small SEM indicates a good absolute reliability of the measure. The Minimal Detectable Change (MDC) was measured using the absolute SEM (MDC = SEM × 1.96 × √2) [26]. The MDC indicates the minimal amount of change that can be confidently interpreted as a real change. A small MDC indicates a more sensitive measurement. All analyses were performed using IBM SPSS Statistics version 28 (SPSS Inc., Chicago, IL, USA) with two-sided tests at α = 0.05 significance level.

## 3. Results

### 3.1. Characteristics of the Included Sample

Of the fifty-five patients identified, seventeen (30.9%) were not interested in participating and one patient (1.8%) had no internet coverage. Therefore, total of 37 (67.3%) patients presenting with gastrointestinal cancer who consented were included. All participants completed both assessments. The median patient age was 54 years old, with most being female (64.9%). Most of the participants presented with colorectal cancer (62.2%), followed by pseudomymoxa peritonei (16.2%). The majority of the participants were familiar with technology, including smartphone/computer (70.3%) and iPad or tablet devices (73.0%). The detailed characteristics of the included participants are presented in Table 1.

### 3.2. Feasibility

One patient was excluded from the study due to not having internet coverage at their place of residence (regional area). Two participants (5.4%) out of the thirty-seven patients that were included in the study presented with internet connectivity issues at the beginning of the test (during the instruction period). These connections were resolved and the remote 5STS tests were able to be completed without further incidents.

### 3.3. Reliability

The face-to-face and remote 5STS assessments were performed 1.3 days apart (range = 1 to 3 days) on average. The mean (SD) time taken to complete the face-to-face and remote 5STS tests was 9.1 (2.4) and 9.5 (2.3) seconds, respectively (*p* < 0.052). The individual 5STS times are illustrated in Figure 1.

The mean difference and upper/lower limits of agreement for the 5STS tests are illustrated on the Bland-Altman plot (Figure 2). There was excellent agreement between face-to-face and remote tests, with only a small proportion of observations falling outside of the limits of agreement.

The reliability of the remote 5STS test was excellent, with an ICC = 0.957 (95%CI = 0.88 to 0.98), *p* < 0.001. The SEM and the MDC were 0.176 and 0.488 s, respectively.

### 3.4. Safety

During the face-to-face and remote assessments, no serious or minor adverse events were observed during testing.

## 4. Discussion

The findings of this study demonstrated that the remote 5STS test is feasible, reliable, and safe in patients undergoing treatment for gastrointestinal cancer. These results support the use of remote assessment in this population, especially if conducting the 5STS test face-to-face would facilitate the assessment of patients’ functional capacity in those living in remote regions or in another state.

The remote 5STS test was considered safe, with no adverse events observed across the study period. During the remote assessments, the study research officer was able to identify a safe place to perform the 5STS test with ease. For safety reasons, all 5STS remote tests were performed with a support person near to the participant; however, their involvement was not required. To the authors knowledge, this is the first study to investigate the safety and reliability of a remote 5STS test in this population. Other studies have investigated psychometrics measurements of the 5STS test in other settings and populations [27]. Similar findings were reported in a study investigating the safety and reliability of the 5STS test in older patients hospitalised in an intensive care unit. Of the 288 face-to-face tests performed (*n* = 96 unique patients), no discontinuation or adverse events were observed [28]. Similarly, in a cancer population, the 30 s STS was found to be safe and feasible when remotely tested on telehealth in 30 patients [29].

Only one patient was not able to be included in the study, as they had no internet coverage at their area of residency. Our hospital is a tertiary/quaternary referral centre in Australia for the treatment of gastrointestinal cancer patients. As such, many of the included patients (40–60%) were from other catchment areas, including regional and interstate areas. Despite the disparity in terms of place of residency, internet coverage was not a major issue during the remote 5STS tests, with only two patients having connectivity issues during the start of the online assessments. These issues were soon resolved, and the test was able to be normally conducted. Similarly, the study conducted by Ogawa et al. also reported a small number of connectivity issues (5%) within their remote physical assessments, including 30 s arm curls, the 30 s STS, and the 2-min step test in older veterans [14]. Thus, it is feasible to perform the remote 5STS test in regions with good internet connectivity.

Early evidence has supported the validity and reliability of a variety of physical assessments remotely performed in other populations [13,14,30,31,32,33]. In our study, the remote 5STS test showed excellent reliability (ICC = 0.957) and no significant systematic errors were observed. The reliability (ICC) in other populations for the remote 5STS was also supportive, including among older veterans (remote assessments performed by two different assessors, ICC = 0.999) [14] and older adults (face-to-face vs. remote assessments, ICC = 0.960) [13]. The 30 s STS, is a variation of the 5STS test, and has also demonstrated similar reliability outcomes in knee osteoarthritis (ICC = 0.920) [30], cancer survivors and carers (ICC = 0.860) [32], and multiple sclerosis (ICC = 0.974) [31]. The SEM and MDC values of the 5STS test were 0.176 and 0.488 s, respectively. These values are of importance for clinicians and can be used to assess the lower limb strength, to determine if the patients condition is improving or deteriorating over time. While the reliability of the remote 5STS test has been evaluated in other populations, mostly in older adults, it was important to determine its feasibility, reliability, and safety in cancer patients. Recent studies have demonstrated a significant reduction in physical capacity and lower limb strength in cancer patients when compared to non-cancer populations [9,34], which is further exacerbated by preoperative neoadjuvant therapy. Therefore, our study was able to evaluate these important psychometric measurements in cancer patients.

Some of the strengths of this study included the powered sample size, the strong methodology utilised (i.e., randomising the first 5STS test to either face-to-face or remote), the clinical importance of having a reliable and safe strength and function measure, and the utilisation of a trained research officer who measured both 5STS tests using a standardised approach. This study has some limitations that need to be addressed. First, this study included patients with gastrointestinal cancer that were mostly tech-savvy and in possession of a computer or mobile device with good internet connectivity and a video camera. Secondly, despite this study not collecting information on the reasons for non-participation, these patients appear to be similar in terms age, gender, type of cancer, and country of birth when compared to the included patients. In this study, 31% of patients approached were not interested in participating. A better understanding of the reasons for non-participation would enhance the evidence on the feasibility of conducting the remote 5STS in this population. If the reasons for non-participation were mostly related to common issues such as time commitment, then it is unlikely to have impacted our results. If, however, these patients had substantially worse physical function or less access to technology, then that could impact the generalisability of our findings. Therefore, our results cannot be generalised to all patients and countries where patients may be less familiar with technology and internet coverage is suboptimal. In addition, as our study included one experienced research officer collecting all 5STS test measures, the interrater reliability was not able to be determined, and represents an important future study alongside measuring validity in this cancer population. Lastly, the required resources, financial advantages and disadvantages, together with the patient experience of completing the 5STS test face-to-face or remotely, should also be investigated in future studies, as these factors could be identified as a barriers to clinicians and/or patients using the remote 5STS test.

## 5. Conclusions

Our study provides important psychometric and clinical utility information on the remote 5STS test conducted among patients with gastrointestinal cancer. The study provides supporting evidence that the remote 5STS test is feasible, reliable, and safe in patients with gastrointestinal cancer and can be used in both clinical and research settings.

## Figures and Tables

**Figure 1 cancers-15-02434-f001:**
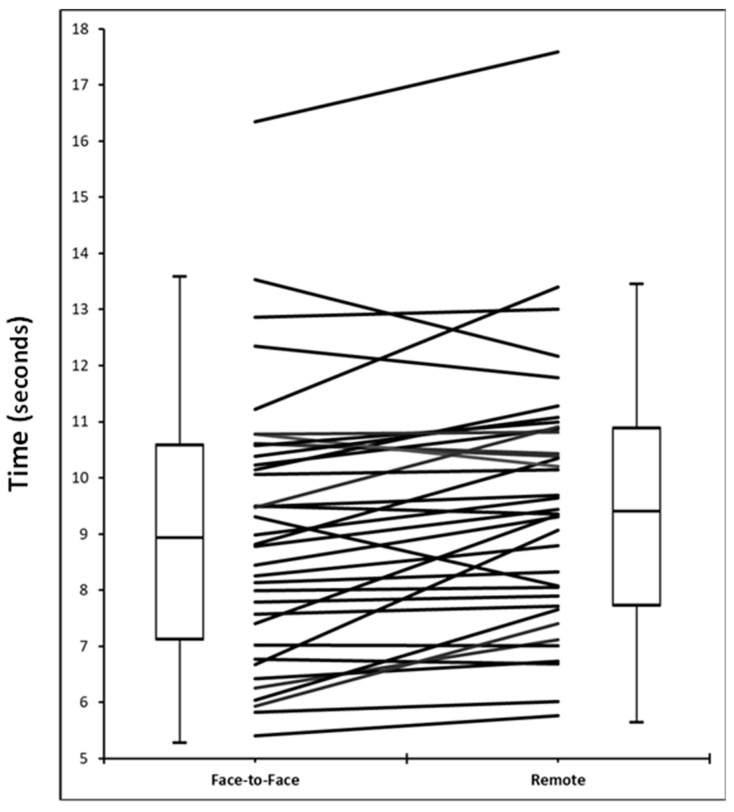
Time taken to complete face-to-face and remote five times sit to stand tests (*n* = 37). Face-to-face: Median (IQR) = 9.0 (7.2 to 10.6) and mean (SD) = 9.1 (2.4); Remote: Median (IQR) = 9.4 (7.8 to 10.9) and mean (SD) = 9.5 (2.3), *p* < 0.058.

**Figure 2 cancers-15-02434-f002:**
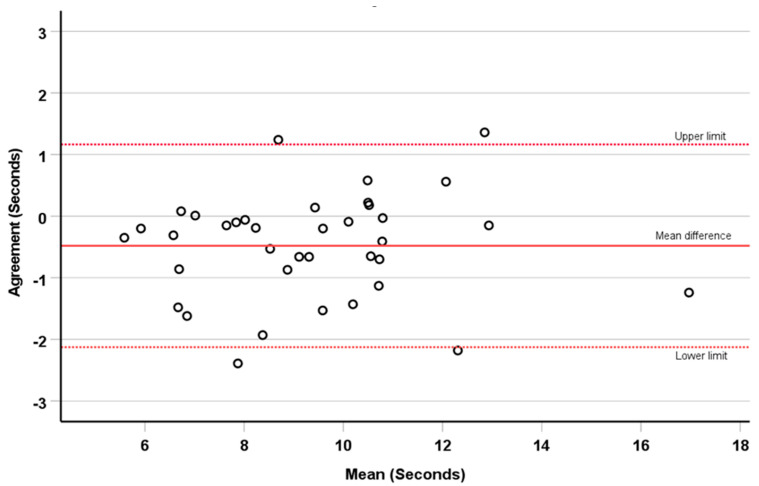
Bland-Altman plot of agreement between face-to-face and remote five times sit to stand tests. The mean difference was −0.48 s (continuous red line) and the upper and lower limits were 1.16 and −2.12 s (broken red lines).

**Table 1 cancers-15-02434-t001:** Characteristics of the included sample (*n* = 37).

Baseline Variables	Frequency (Percentage) or Median (Interquartile Range)
Age, years	54.0 (46.0 to 61.5)
Gender, female	24 (64.9%)
Body mass index, kg/m^2^	24.1 (22.7 to 28.4)
Country of birth	
Australia	25 (67.6%)
Overseas	12 (32.4%)
Language spoken at home	
English	31 (83.8%)
Other	6 (16.2%)
Caring responsibilities	13 (35.1%)
Level of education	
Primary school—Year 12	13 (35.1%)
Technical certificate or diploma	7 (18.9%)
University degree	17 (46.0%)
Employment status	
Full-time/Part-time	23 (62.2%)
Retired/sick leave	12 (32.4%)
Unemployed	2 (5.4%)
Type of cancer	
Anal	2 (5.4%)
Appendix	3 (8.1%)
Colorectal	23 (62.2%)
Gastrointestinal Stromal Tumour	1 (2.7%)
Pseudomyxoma Peritonei	6 (16.2%)
Retroperitoneal Liposarcoma	1 (2.7%)
Small Bowel Adenocarcinoma	1 (2.7%)
Familiarity with technology	
Smartphone or computer	
Very familiar	26 (70.3%)
Familiar	9 (24.3%)
Not at all familiar	2 (5.4%)
iPad or tablet device	
Very familiar	27 (73.0%)
Familiar	8 (21.6%)
Not at all familiar	2 (5.4%)
Numerical pain rating score ^a^	1.0 (0.0 to 3.0)
Distress thermometer ^b^	1.0 (0.0 to 2.5)
Fatigue Severity Scale ^c^	26.0 (14.0 to 38.5)
Meeting WHO physical activity recommendations ^d^	
Yes	15 (40.5%)
No	22 (59.5%)

^a^ Pain scores range from 0 to 10, where 10 indicates the worst pain; ^b^ Distress scores range from 0 to 10, with higher scores indicating higher distress; ^c^ Fatigue scores range from 9 to 63, with higher scores indicating more severe fatigue; ^d^ Meeting the World Health Organisation physical activity recommendations; WHO = World Health Organisation.

## Data Availability

The data presented in this study are available on request from the corresponding author.

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
