# Peer review of "Feasibility, Reliability, and Safety of Remote Five Times Sit to Stand Test in Patients with Gastrointestinal Cancer"

_cancers, 2023, doi:10.3390/cancers15092434_

Round 1
Reviewer 1 Report
This study investigated the feasibility and reliability of remote five times sit to stand test in surgical patients. I have some suggestions listed below.
1. Please mention the advantage and disadvantages of finicial abalysis between two methods. Although remote method eliminate the possibility of being infected by COVID, it needs extra resources to provide this service.
2. As to the 5STS, I wonder if this test is validated to have a statistical relationship with surgical outcomes or not.
Author Response
REVIEWER 1:
Point 1. This study investigated the feasibility and reliability of remote five times sit to stand test in surgical patients. I have some suggestions listed below.
Authors’ responses: We thank reviewer #1 for their suggestions.
----------------------------------------------------------------------------------------
Point 2. Please mention the advantage and disadvantages of finicial abalysis between two methods. Although remote method eliminate the possibility of being infected by COVID, it needs extra resources to provide this service.
Authors’ responses: We agree with the reviewer that future studies investigating the advantages and disadvantages of completing the 5STS test face-to-face or remotely should be investigated. We have made changes to the discussion section to cover this matter.
Page 9: “Lastly, the required resources, financial advantages and disadvantages, together with patient experience of completing the 5STS test face-to-face or remotely should also be investigated in future studies, as these factors could be identified as a barrier to clinicians and patients.”
----------------------------------------------------------------------------------------
Point 3. As to the 5STS, I wonder if this test is validated to have a statistical relationship with surgical outcomes or not.
Authors’ responses: This is an important point. While this study have not investigated the relationship of 5STS test with surgical outcomes, our group have recently published on the association between preoperative 5STS test with length of hospital stay and postoperative complications. However, the association between the 5STS test assessed remotely with postoperative surgical outcomes is currently unknown.
Reference: Preet G. S. Makker, Cherry E. Koh, Nabila Ansari, Nicole Gonzaga, Jenna Bartyn, Michael Solomon, Daniel Steffens. Functional outcomes of cytoreductive surgery and hyperthermic intraperitoneal chemotherapy - a prospective cohort study. Annals of Surgical Oncology. 2022 October. DOI: 10.1245/s10434-022-12691-x
Reviewer 2 Report
This interesting work suggests the feasibility of doing a remote sit to stand x 5 test in research settings.
1. The work seems very preliminary, and perhaps not enough critical mass of data for a high impact journal.
2. The conclusion that this is "feasible" is intuitive, in the sense that surely some patients can do this. However, the authors ignore the fact that about 1/3 of their patients weren't included in the study because they didn't want to participate or had teleconnection issues. Doesn't this mean that the technique is really not feasible over a broad range of research subjects?
3. Of the "excluded" patients, most were unwilling to participate. To what extent could that have been because they might have felt uncomfortable with the physical demands of the test. Could that have biased the study results? Could such patients have had more marginal results that might not have correlated as well? Could such patients have had a higher chance of falling down or some other adverse event?
4. The patient population seems relatively young and did relatively well in this test. Both for research and clinical purposes, isn't the test more important for older frailer patients?
Author Response
REVIEWER 2:
Point 1. This interesting work suggests the feasibility of doing a remote sit to stand x 5 test in research settings.
Authors’ responses: We thank reviewer #1 for their suggestions.
----------------------------------------------------------------------------------------
Point 2. The work seems very preliminary, and perhaps not enough critical mass of data for a high impact journal.
Authors’ responses: We thank reviewer #2 for their comment. We would like to emphasise that this study was performed using appropriate scientific design and adequately powered to answer whether the remote 5STS is feasible, reliable, and safe. We also believe this manuscript will be widely cited, as this is the first study to validate the 5STS test within a cancer population. The information provided in this manuscript will strengthening the body of evidence in this area.
----------------------------------------------------------------------------------------
Point 3. The conclusion that this is "feasible" is intuitive, in the sense that surely some patients can do this. However, the authors ignore the fact that about 1/3 of their patients weren't included in the study because they didn't want to participate or had teleconnection issues. Doesn't this mean that the technique is really not feasible over a broad range of research subjects?
Authors’ responses: We thank the reviewer #2 for their comments. We agree that patient recruitment is a challenge in clinical research with most studies struggling to even reach their planned sample size. A total of 31% of the patients approached were not interested in participating - unfortunately the reasons for no participation is unknow and we are not able to comment on this matter. When we compared our recruitment rate with previous studies, the first thing that we identified was that most of the studies are not clear on the amount of patients screened and this is related to the low quality of the current literature in this field. One study conducted by Melo 2019, described that of 231 eligible patients identified, 118 refused to participate (51%). This indicates the challenges of recruiting patient to clinical studies.
We have made changes to our discussion section to address this matter.
Page 9. “…Secondly, we haven’t collected information on the reasons for non-participation. In this study, 31% of patients approached were not interested in participating. The reasons for non-participation would enhance the evidence on the feasibility of conducting the remote 5STS…”
----------------------------------------------------------------------------------------
Point 4. Of the "excluded" patients, most were unwilling to participate. To what extent could that have been because they might have felt uncomfortable with the physical demands of the test. Could that have biased the study results? Could such patients have had more marginal results that might not have correlated as well? Could such patients have had a higher chance of falling down or some other adverse event?
Authors’ responses: We thank reviewer #1 for their suggestions. We agree that understanding the reasons of no participation in clinical research is important. Unfortunately, this information was not sought in the current study. Overall, recruitment of patients to clinical research is a challenge. The 5STS test is a very simple and safe test to be performed. No issues were observed within the patients that agree to participate.
We have made changes to our discussion section to address this matter.
Page 9. “…Secondly, we haven’t collected information on the reasons for non-participation. In this study, 31% of patients approached were not interested in participating. The reasons for non-participation would enhance the evidence on the feasibility of conducting the remote 5STS…”
----------------------------------------------------------------------------------------
Point 5. The patient population seems relatively young and did relatively well in this test. Both for research and clinical purposes, isn't the test more important for older frailer patients?
Authors’ responses: We thank reviewers #1 for their comment. The characteristics of the sample is based on the sampled patients. The 5STS test has demonstrated important associations with postoperative complications and length of hospital stay in patients undergoing cancer surgery. This information is relevant during the preoperative period, so patients physical function could be optimised before surgery.
Reference: Preet G. S. Makker, Cherry E. Koh, Nabila Ansari, Nicole Gonzaga, Jenna Bartyn, Michael Solomon, Daniel Steffens. Functional outcomes of cytoreductive surgery and hyperthermic intraperitoneal chemotherapy - a prospective cohort study. Annals of Surgical Oncology. 2022 October. DOI: 10.1245/s10434-022-12691-x
Round 2
Reviewer 1 Report
My comments had been addressed comprehensively.
Reviewer 2 Report
The authors have simply responded by disagreeing with my concerns.
I still believe that these results are severely limited by the >30% non participation rate and too preliminary for publication in a high impact journal.